# The Contribution of Negative Expectancies to Emotional Resilience

**DOI:** 10.3390/bs15040531

**Published:** 2025-04-15

**Authors:** James Tough, Ben Grafton, Colin MacLeod, Bram Van Bockstaele

**Affiliations:** 1Centre for the Advancement of Research on Emotion, School of Psychological Science, University of Western Australia, 35 Stirling Highway, Crawley, WA 6009, Australia; ben.grafton@uwa.edu.au (B.G.); colin.macleod@uwa.edu.au (C.M.); bram.vanbockstaele@ugent.be (B.V.B.); 2Department of Developmental, Personality, and Social Psychology, Ghent University, Henri Dunantlaan 2, 9000 Gent, Belgium

**Keywords:** cognition, expectancy, attention, anxiety, resilience

## Abstract

Anxiety reactivity, i.e., the degree to which state anxiety becomes elevated, has been used as a measure of emotional resilience in anticipation of potentially stressful events and has been found to correlate with elevations in event-related negative expectancy bias. The present study aimed to replicate this finding and investigate whether negative expectancy bias is also associated with low emotional resilience in the wake of the event, measured as anxiety perseveration, i.e., the degree to which state anxiety remains elevated after the event. A sample of undergraduate students was informed they would watch a film montage and presented with the choice to access negative or benign information relevant to the film montage. They were asked to rate their negative expectancy bias and state anxiety both before and after accessing this information, which permitted a measure of anxiety reactivity and negative expectancy bias elevation. Participants then watched the film montage and rated their experience and state anxiety again, which allowed for a measure of anxiety perseveration. The results revealed that negative expectancy bias predicted anxiety reactivity and predicted anxiety perseveration indirectly through its impact on the perceived negativity of the event. Although further investigation is required, these findings suggest interventions targeting negative expectancy bias may improve emotional resilience both in anticipation of and in the wake of stressful events.

## 1. Introduction

### 1.1. The Importance of Emotional Resilience

Emotional resilience has been defined as the ability to bounce back from high levels of negative emotion in the face of adversity ([12]; [5]). Compared to people with lower levels of emotional resilience, people with higher levels of emotional resilience are more socially connected, enjoy better academic and employment outcomes, are more physically healthy, and demonstrate better mental health outcomes ([11]). As such, it is important to advance understanding of the mechanisms that underpin variation in emotional resilience, as doing so carries the prospect of enhancing well-being across the population and in targeted populations such as those diagnosed with psychopathology.

### 1.2. Anxiety Reactivity and Perseveration as Manifestations of Emotional Resilience

Research has consistently demonstrated a relationship between emotional resilience and negative emotion ([14]; [13]), particularly state anxiety, across a range of stressful experiences. In these studies, it has been common to differentiate between state anxiety experienced both before and after a potentially stressful event. Specifically, the degree to which state anxiety becomes elevated as potentially stressful events approach ([7]) and the persistence of state anxiety after the event ([17]; [2]) are both recognized as key indicators of emotional resilience. As such, both the elevation and perseveration of state anxiety reflect anxiety-focused measures of emotional resilience in the lead up to a potentially stressful event and emotional resilience in the aftermath of the event, respectively. This contrast echoes the distinction between anxiety reactivity and anxiety perseveration, which were identified as distinct dimensions of anxiety by [18] ([18]), whereby anxiety reactivity may reflect elevations in state anxiety as a potentially stressful event approaches, and anxiety perseveration may reflect state anxiety following a potentially stressful event. Given this, the present study refers to anxiety reactivity as a manifestation of emotional resilience in the lead up to a potentially stressful event and to anxiety perseveration as a manifestation of emotional resilience following a potentially stressful event. The identification of these two manifestations of emotional resilience to state anxiety generates two further questions, which concern the nature of the cognitive mechanisms that underly variation in anxiety reactivity and whether these same mechanisms contribute to variation in anxiety perseveration.

### 1.3. Negative Expectancy Bias and Anxiety Reactivity

[23] ([23]; Preprint) addressed one of these questions pertaining to the cognitive mechanisms underlying variation in anxiety reactivity by testing two hypotheses derived from insights from cognitive psychology suggesting that negative expectancies about a potentially stressful event may contribute to variation in anxiety reactivity in anticipation of that event. Specifically, Tough et al. investigated (a) whether the degree to which negative rather than positive expectancies concerning a potentially stressful event become elevated as the event approaches contributes to anxiety reactivity and (b) whether the degree to which negative rather than positive expectancies become elevated is driven by negative selective interrogation bias or the degree to which individuals selectively access more negative, compared to positive, information about that event. To achieve this, they employed a paradigm whereby participants were informed they would watch a (fictitious) film montage showcasing the experience of refugees, which may at times be both uplifting and distressing. Participants were asked to rate their negative expectancy bias, i.e., the degree to which they held a greater proportion of negative compared to positive expectancies, and state anxiety at two time points, which allowed for a measure of the degree to which both negative expectancy bias and state anxiety became elevated as the film montage viewing approached. Participants were also given the opportunity to access either negative or benign information statements about the film montage in between the two assessment points. Tough et al. found, as predicted, that the resulting elevation in negative expectancies directly predicted greater anxiety reactivity, as indicated by greater elevations in state anxiety, in anticipation of the approaching event. This pattern of findings has also been established in other research in cognitive psychology. For example, [23] ([23]; Preprint) observed a positive association between negative expectancy bias elevation and changes in state anxiety, and [15] ([15]) observed that negative expectancy bias indirectly predicts negative affect. Indeed, these recent findings echo the results of previous research, which has consistently linked negative expectancies to elevated anxiety ([20]; [3]) Critically, these findings highlight the protective value of decreased negative expectancies against anxiety reactivity and, by association, emotional resilience in the lead up to a potentially stressful event. Interestingly, Tough et al. also found that variation in the degree to which individuals formed negative expectancies about the event was predicted by variation in the tendency to selectively access more negative than positive information relevant to the event when given a choice between negative information and benign information.

### 1.4. Negative Expectancy Bias and Anxiety Perseveration

Although this research has contributed to our understanding of the mechanisms underlying emotional resilience in the lead up to a potentially stressful event, it remains unclear whether those same mechanisms also contribute to variation in emotional resilience in the aftermath of the event. While anxiety perseveration may also be driven by the degree to which individuals expect potentially stressful events to be more negative compared to positive, it is possible that it may be driven by different mechanisms. If it were the case that negative expectancy bias immediately before the event also predicts anxiety perseveration, it would be important to distinguish between two competing possibilities concerning the nature of this association. One possibility is that individual differences in negative expectancy bias directly contribute to variance in anxiety perseveration. Alternatively, it may be the case that individual differences in negative expectancy bias indirectly predict variance in anxiety perseveration through the mediating influence of the degree to which individuals reported experiencing the stressor as more negative than positive. This notion is supported by previous research finding that violations of expectations can alter psychopathology ([16]).

### 1.5. Present Study

Considering previous research, the present study has two key functions, the first of which is to replicate the findings of [23] ([23]; Preprint) concerning the predictive validity of negative expectancy bias for anxiety reactivity. This is critical given the relatively sparse research addressing the relationship between negative expectancy bias and anxiety reactivity, which necessitates further empirical support ([6]). Secondly, this study aims to investigate whether negative expectancy bias also plays a role in driving variation in anxiety perseveration and, if so, to determine whether this reflects a direct or indirect association. Investigating the role of negative expectancy bias in anxiety perseveration is imperative, as a positive association could indicate that targeting expectancies in clinical interventions may be protective against not only the elevation of anxiety but also its perseveration.

To address these two issues, the present study adopted the methodology employed by [23] ([23]; Preprint), which satisfies the criteria necessary to test whether negative expectancy bias is associated with anxiety reactivity. To investigate whether negative expectancy bias also predicts anxiety preservation, either directly or indirectly, the present study required a measure of anxiety perseveration. To achieve this, participants were exposed to a potentially stressful event, and their state anxiety was assessed following this event so that a measure of anxiety perseveration could be obtained. Additionally, a measure of the degree to which individuals reported experiencing the event as more negative than positive was obtained to determine the nature of the hypothesized association between negative expectancy bias and anxiety perseveration.

## 2. Method

### 2.1. Participants

A total of 78 (52 self-reported female, 25 self-reported male, and 1 self-reported non-binary) students were recruited from the University of Western Australia psychology undergraduate population. Participants received course credits in exchange for partaking in the study. The mean age of participants was 22.27 (*SD* = 8.76; range = 18–53). 

### 2.2. Film Montage

Participants were informed they would view a film montage that conveys both emotionally negative and benign content. The present study required that participants experienced this event so that a measure of anxiety perseveration following the event could be obtained. To achieve this, a film montage was created using film clips taken from popular media. Eight film clips were used in total, of which four conveyed emotionally negative content, and four conveyed emotionally benign content. These film clips were independently rated on valence and vividness, demonstrating that the negative and benign film clips differed significantly across both valence (*t*(5) = 25, *p* < 0.001) and vividness (*t*(5) = 8.78, *p* < 0.001). Each video clip was 60 s in duration, resulting in a total duration of twelve minutes for the film montage. The videos were presented in a random order, with no more than three videos in a row conveying content with the same valence.

### 2.3. Questionnaires

#### 2.3.1. Negative Expectancy Bias Assessment

The present study utilized the 8-item questionnaire originally developed by [22] ([22]) to assess whether participants held more negative, compared to positive, expectancies about the film montage. Four of these items were negative expectancy items, which described negative emotions that generally indicate an unpleasant experience, such as “appalled”. The remaining four items were positive expectancy items, which described positive emotions that generally indicate a pleasant experience, such as “uplifted”. Participants were asked to rate how likely they expected they were to experience each negative expectancy item and positive expectancy item during the film montage. This expectancy assessment was delivered using a computer, where each item was displayed in the center of the screen. A sliding Likert scale, with scores ranging from 1 to 100 (labelled “not at all” and “very much”, respectively) was positioned below each item. Participants were asked to drag the marker to the position that most accurately reflected their expected likelihood of experiencing each emotion while watching the film montage. Scores on these items ranged from 1 (low expectancy) to 100 (high expectancy).

To compute a measure of negative expectancy bias, the mean score of positive expectancy items was deducted from the mean score of negative expectancy items. The resultant score reflects the relative endorsement of negative, compared to positive, expectancies. A positive bias score signified a greater tendency to expect negative than positive emotions, while a negative score indicated a greater tendency to expect positive rather than negative emotions. A bias score of zero suggests an equal endorsement of both negative and positive expectancies.

#### 2.3.2. Negative Experience Index Assessment

We also measured the degree to which participants experienced the film montage as negative rather than positive to address the second goal of the study. To achieve this, a measure reflecting the degree to which participants found that viewing the film montage was more negative than positive was created using the items from the Negative Expectancy Bias Assessment ([22]; see Appendix A). As such, participants were asked to rate the degree to which each item aligned with their emotional experience while viewing the film montage. This allowed for comparison between how participants expected to feel and how they felt after the film montage. The experience assessment was delivered and rated in the same manner as the expectancy assessment. As such, item scores ranged from 1 (“not at all”) to 100 (“very much”). A negative experience index was computed by subtracting each participants’ average score on the positive experience items from their average score on the negative experience items. A negative experience index with a positive score indicated that, on average, the participants more strongly endorsed negative experience items compared to positive experience items, whereas a negative score indicated that, on average, the participants more strongly endorsed positive experience items compared to negative experience items. As above, a score of zero indicated that participants endorsed both negative experience items and positive experience items with approximately equal strength.

#### 2.3.3. Assessment of Anxiety Reactivity and Perseveration

In the present study, state anxiety was assessed using an amended version of the STAI-6 ([19]) used by [22] ([22]). The STAI-6 has high convergent and discriminant validity and internal consistency reliability ([19]). Participants were asked to respond to six items that represent common symptoms associated with state anxiety, such as “I feel tense”, by indicating the extent to which each item currently resonated with them. Participants responded to each item with a 100-point Likert scale, where a score of 1 indicated “not at all” and a score of 100 indicated “very much”. State anxiety scores were calculated for each participant as sum scores across each item, where a higher score indicated greater state anxiety.

The STAI-6 was administered to assess state anxiety at multiple points to provide measures of anxiety reactivity and anxiety perseveration. To measure anxiety reactivity, state anxiety was assessed twice: once at the beginning of the experiment and again immediately before viewing the film montage. The duration between the two state anxiety assessments was approximately 20 min but varied among participants. To calculate an index of anxiety reactivity, the baseline state anxiety score was subtracted from the pre-film montage state anxiety score, where higher scores indicated greater anxiety reactivity. Furthermore, post-film montage state anxiety was measured, and these scores served as an indicator of anxiety perseveration.

#### 2.3.4. Information Statements Describing Film Montage

To provide participants with relevant information about the film montage, two distinct information sets were created encompassing 32 statements. These statements detailed negative or benign content associated with each of the eight video clips. These statements were presented as purported feedback from prior viewers. Specifically, each of the four negative video clips was paired with four negative information statements, while each of the four benign video clips was paired with four benign information statements. This resulted in a total of 16 negative information statements, each describing a negative scene and a supposed viewer’s negative emotional reaction (e.g., “It was upsetting to hear the panicked and anguished shouts of the others when their friend died”). The remaining 16 benign information statements described benign scenes and corresponding benign emotional reactions (e.g., “It was truly impressive to see how the leader inspired such loyalty and devotion in his soldiers”). To ensure distinctiveness, no two statements referenced the same video clip excerpt or utilized identical adjectives to describe emotional responses. Statement lengths were standardized, ranging from approximately 90 to 110 characters.

### 2.4. Information Statement Presentation and Selective Interrogation Bias Assessment

We used the paradigm used by [23] ([23]; Preprint) to present the negative information statements and benign information statements and assess participants’ negative expectancy bias, anxiety reactivity, and negative selective interrogation bias. In each of the 32 trials of the task, participants were presented with one of the negative information statements and one of the benign information statements for 8000 ms.

The beginning of each trial was marked by the presentation of a pair of squares, of which one was blue, and one was yellow, containing either a plus or a minus symbol inside them. These symbols corresponded to the valence of the information that could be accessed later in the trial. For half of the participants, as displayed in Figure 1, the blue square contained a plus symbol, which corresponded to a benign information statement, and the yellow square contained a minus symbol, which corresponded to a negative information statement. The reverse was true for the other half of the participants. This pair of squares appeared for 2000 ms. Following this, the same pair of squares was presented again without the plus or minus symbol and inside a white box, positioned in the locations in which the statements would appear if accessed. As in the example in Figure 1, the white box with the blue square indicated the location in which the benign information statement would appear if accessed, and the white box with the yellow square indicated the location in which the negative information statement would appear if accessed. The location of the pair of statements was either horizontally aligned, where one statement appeared in the left half of the screen and the other in the right half of the screen, or vertically aligned, where one statement appeared in the top half of the screen and the other in the bottom half of the screen. These conditions occurred with equal frequency, and the location in which the negative information statements and benign information statements appeared was equally balanced between all four possible locations.

Participants were asked to select which of the two statements they wished to see by entering a single key on the number pad corresponding to the location of the statement. For example, if a participant wanted to access a benign information statement, indicated by a white box containing a blue square in the left half of the screen, the participant would enter the “left key” on the number pad, indicated by the left-facing arrow. Similarly, if the participant wanted to access a negative information statement, indicated by a white box containing a yellow square in the right half of the screen, the participant would enter the “right key” on the number pad, indicated by the right-facing arrow. Following the registration of the participants’ response, the selected statement appeared in the appropriate location for 8000 ms.

A negative selective interrogation bias index was calculated, representing the proportion of total trials in which participants chose to access negative information statements. This index ranged from 0 to 1, where a score of 1 indicated exclusive selection of negative information statements, and a score of 0 indicated minimal or no selection of negative information statements.

### 2.5. Procedure

After obtaining informed consent and demographic information, participants were briefed on the film montage, which they were told might contain both positive and negative content. They then completed initial assessments of expectancies and state anxiety. Participants received instructions and practiced the Information Statement Presentation and Selective Interrogation Bias Assessment using neutral statements. This assessment yielded a negative selective interrogation bias score. Subsequent expectancies and state anxiety ratings allowed for the calculation of negative expectancy bias elevation and anxiety reactivity. Negative expectancy bias elevation was determined by subtracting initial negative expectancy bias scores from pre-film montage scores, with higher scores indicating increased negativity. Participants then viewed the film montage, followed by post-film ratings of their experience and state anxiety, which were used as indicators for the negative experience index and anxiety perseveration. This concluded the in-person testing. Participants received a written debriefing and were thanked. This procedure is detailed in Figure 2. The study was approved by the Human Resource Ethics Office (RA/4/1/5243).

## 3. Results

### 3.1. Data Cleaning, Preparation, and Analysis

Of the seventy-eight participants, one was removed from data analysis due to an error that resulted in incomplete data for the outcome variables. An outlier analysis was conducted following [8]’s ([8]) recommendations, which define outliers as scores that fall outside three times the inter-quartile range from either the lower or upper quartile. Zero outlying scores were removed. As such, the analytical sample comprised 77 participants.

### 3.2. Descriptive Statistics

A summary of the descriptive statistics is outlined in Table 1. The assumption of normality, as indicated by the Shapiro–Wilk test for normality, was violated for anxiety reactivity (*W* = 0.951, *p* = 0.005), negative expectancy bias elevation (*W* = 0.963, *p* = 0.026), and negative selective interrogation bias (*W* = 0.947, *p* = 0.003). Thus, non-parametric correlations were used to investigate these effects for variables with non-normal distributions.

### 3.3. Is There an Association Between Negative Expectancy Bias and Anxiety Reactivity?

As per the first aim of the present study, the following series of analyses were conducted to investigate whether each of the original effects in [23] ([23]; Preprint) concerning the impact of negative expectancy bias on anxiety reactivity were replicated.

#### 3.3.1. Effect Concerning Association Between Negative Selective Interrogation Bias and Anxiety Reactivity

To address the expected positive association between negative selective interrogation bias scores and anxiety reactivity scores, a Spearman correlation was conducted. There was no significant association between negative selective interrogation bias scores and anxiety reactivity scores (ρ = 0.104, *p* = 0.366).

#### 3.3.2. Effect Concerning Association Between Negative Expectancy Bias Elevation and Anxiety Reactivity

To address the expected positive association between negative expectancy bias elevation and anxiety reactivity, a Spearman correlation was conducted. Negative expectancy bias elevation scores were significantly and positively associated with anxiety reactivity scores (ρ = 0.265, *p* = 0.02).

#### 3.3.3. Effect Concerning Association Between Negative Selective Interrogation Bias and Negative Expectancy Bias Elevation

To address the expected positive association between negative selective interrogation bias and negative expectancy bias elevation, we conducted a Spearman correlation. Negative selective interrogation bias scores shared a significant positive association with negative expectancy bias elevation scores (ρ = 0.316, *p* = 0.005).

#### 3.3.4. Effect Concerning Mediation of Association Between Negative Selective Interrogation Bias and Anxiety Reactivity by Negative Expectancy Bias Elevation

To address whether, as expected, negative expectancy bias elevation fully mediated the relationship between negative selective interrogation bias and anxiety reactivity, a mediation analysis was conducted using the medmod package in Jamovi ([21]). As mediation analyses using bootstrapping procedures that are robust to deviations from normality ([1]), no data transformation was conducted prior to this analysis. In the mediation analysis, “negative selective interrogation bias” was entered as the predictor, “anxiety reactivity” was entered as the dependent variable, and “negative expectancy bias elevation” was entered as the mediator.

The results of the mediation analysis revealed variance in negative selective interrogation bias scores did statistically predict variance in negative expectancy bias elevation scores (a path; *B* = 114.04, *p* = 0.005), which in turn predicted variance in the index of anxiety reactivity (b path; *B* = 0.415, *p* = 0.002). Variance in negative selective interrogation bias scores did statistically predict variance in the index of anxiety reactivity in a manner that is mediated by variance in negative expectancy bias elevation scores (ab path; *B* = 47.3, *p* = 0.04). Standard estimation procedures were used in Jamovi to test the significance of the indirect effect, and 95% confidence intervals were computed by determining the indirect effects at the 2.5th and 97.5th percentiles (95% CI [2.26, 92.4]). The indirect effect was statistically significant, as the confidence intervals did not include zero. The index of negative expectancy bias elevation fully mediated the relationship between negative selective interrogation bias and the index of anxiety reactivity, as the total effect (c path; *B* = 23.1, *p* = 0.655) and direct effect (c’ path; *B* = 51.3, *p* = 0.636) were both non-significant. This indicates that the index of negative selective interrogation bias may influence the index of anxiety reactivity, potentially through its association with the index of negative expectancy bias elevation.

### 3.4. Is There an Association Between Negative Expectancy Bias and Anxiety Perseveration?

As per the second aim of the current study, the following analyses were conducted to investigate whether there is an association between negative expectancy bias and anxiety perseveration and, if so, whether this reflects a direct effect or an indirect effect mediated by negative experience index.

#### 3.4.1. Did Negative Expectancy Bias Predict Anxiety Perseveration?

To test the prediction that greater negative expectancy bias scores immediately before the event are associated with greater anxiety perseveration, a Pearson correlation was conducted. Negative expectancy bias scores were found to be positively associated with anxiety perseveration scores (*r* = 0.509, *p* < 0.001).

#### 3.4.2. Does Negative Expectancy Bias Predict Anxiety Perseveration Directly or Through the Mediating Influence of Negative Experience Index?

To discriminate the validity of the two competing predictions concerning the nature of the relationship between negative expectancy bias and anxiety perseveration, a mediation analysis was conducted using the medmod package in Jamovi ([21]). In the mediation analysis, “negative expectancy bias” immediately before the film montage was input as the predictor, “negative experience index” was input as the mediator, and “anxiety perseveration” was input as the dependent variable.

The results of the mediation analysis revealed variance in negative expectancy bias scores did statistically predict variance in negative experience index scores (a path; *B* = 0.618, *p* < 0.001), which in turn predicted variance in anxiety perseveration scores (b path; *B* = 0.732, *p* < 0.001). Variance in negative expectancy bias scores statistically predicted variance in anxiety perseveration scores in a manner mediated by variance in negative experience index scores (ab path; *B* = 0.452, *p* < 0.001). Standard estimation procedures were used in Jamovi to test the significance of the indirect effect, and 95% confidence intervals were computed by determining the indirect effects at the 2.5th and 97.5th percentiles (95% CI [0.22, 0.68]). The indirect effect was statistically significant, as the confidence intervals did not include zero. Negative experience index scores fully mediated the relationship between negative expectancy bias scores and anxiety perseveration scores, as the total effect (c path; *B* = 0.582, *p* < 0.001) was significant, and the direct effect (c’ path; *B* = 0.130, *p* = 0.256) was non-significant. This pattern of results is consistent with the possibility that negative expectancy bias scores influence anxiety perseveration scores through their association with negative experience index scores.

## 4. Discussion

### 4.1. Summary of Findings

The present study aimed to replicate the findings of [23] ([23]; Preprint) concerning the positive association between negative expectancy bias and anxiety reactivity in the context of an approaching potentially stressful event. This study also aimed to investigate whether negative expectancy bias also predicts anxiety perseveration in the wake of a potentially stressful event and, if so, whether this reflects a direct association or an indirect association subject to the mediating influence of how negative the event was perceived to be.

As predicted, all the original effects reported in [23] ([23]; Preprint) were replicated in the present study, except for one. Participants who reported greater elevations in negative than positive expectancies about the film montage tended to also report greater anxiety reactivity, which is consistent not only with [23] ([23]; Preprint) but also with recent research investigating the role of expectancies in predicting anticipatory anxiety ([22]; [15]). Critically, this finding adds further evidence supporting the notion that expectancies may offer protective value against low emotional resilience, manifested as high anxiety reactivity. It was also noted that the degree to which individuals expected the film montage to be more negative than positive was influenced by the degree to which they selectively accessed more negative than positive information, which was also consistent with [23] ([23]; Preprint). However, the tendency to selectively access more negative than positive information was not associated with greater anxiety reactivity, which contradicts the findings of [23] ([23]; Preprint). This discrepancy may be due to the information statements in the present study alluding to multiple unrelated film clips such that abrupt shifts between thematically distinct content may have dampened the elevation of anxiety caused by information relevant to a specific film. Despite this contradiction, the significance of the mediation model in the present study highlights the same overall pattern of findings, whereby negative expectancy bias, which is influenced by which information about the event is accessed, predicts anxiety reactivity, adding to the robustness of [23]’s ([23]; Preprint) findings.

Of relevance to the second aim of the present study, the findings revealed that the degree to which individuals expected the film montage to be more negative than positive predicted the degree to which their anxiety persevered once the event had ended. Importantly, this novel finding suggests that the protective benefits afforded by expectancies to anxiety reactivity are also manifested in reductions in the perseveration of anxiety. Specifically, these findings indicate that individuals who expect a potentially stressful event to be more negative than positive tend to report greater state anxiety in the wake of that event. However, the findings also revealed that the protective benefits of expectancies only influence anxiety perseveration through their impact on the perceived negativity of the event after it has been experienced. In other words, while expectancies may influence the degree to which individuals report experiencing the event as negative, which determines the degree to which anxiety perseveres after the event, they do not directly contribute to variance in anxiety perseveration.

### 4.2. Theoretical and Clinical Implications

Together, these findings have implications for both our theoretical understanding of the cognitive mechanisms underpinning emotional resilience and clinical practice aimed at improving emotional resilience. Firstly, while [18] ([18]) found that anxiety reactivity and anxiety perseveration are psychometrically distinct processes, the findings of this study suggest that they share a common predictor in expectancies. Critically, this highlights the relevance of pre-event cognitive processes in predicting both immediate and sustained responses to stressors, the consideration of which would enhance current models of emotional resilience. In addition to expanding our understanding of the cognitive basis of two dimensions of emotional resilience, this observation generates additional questions that future studies may wish to address, such as whether the protective benefits of expectancies are stronger for anxiety reactivity or perseveration. Secondly, the protective value afforded to both anxiety reactivity and perseveration by expectancies indicates that interventions aimed at targeting negative expectancy bias may be more beneficial than previously suggested by [22] ([22]) and [23] ([23]; Preprint). Negative expectancy bias modification would integrate with existing cognitive–behavioral interventions ([4]) and allow for the modification of negative thoughts specifically concerning the impact of expectancies on both anticipatory and post-event anxiety as well as a reduction in behaviors that maintain negative expectancies. Additionally, changes in negative expectancy bias could serve as a potential marker of treatment efficacy for interventions targeting emotional resilience or anxiety.

It is also important to investigate alternative explanations for these findings before implementing intervention strategies. One interpretation of these findings is that interventions aimed at reducing anxiety perseveration by targeting negative expectancy bias may be successful through their capacity to reduce the perceived negativity of the event when it is experienced. However, the perceived negativity of the event may be influenced by factors other than the degree to which individuals expected it to be negative or even by the discrepancy between their expectancies and their experience. This echoes previous research that found that greater expectancy violations can reduce threat ratings in extinction learning ([10]). Future studies may wish to discriminate other pathways through which the perceived negativity of the event may become elevated in a manner that reduces anxiety perseveration to determine whether modifying expectancies is still a valid intervention target. In such a case, these findings may further incentivize the use of strategies that target negative expectancy bias as a means of improving both anxiety reactivity and perseveration in pathologically anxious individuals.

### 4.3. Limitations and Future Directions

It is also important to consider the limitations of the current study. One limitation is that the information relevant to the film montage was delivered as a compressed package across a brief period of time (see also [23]; Preprint). While this may elicit state anxiety fluctuations ([9]), it is not representative of all naturally occurring stressors, in which individuals often have access to information across extended periods of time. For example, an individual preparing for a cinema screening of a potentially stressful movie might search ratings or discuss the film with others who have seen it across several days in advance, which may impact either the generation of expectancies or the protective value of expectancies against low pre-event emotional resilience. Future studies should aim to assess whether prolonged exposure to information results in variation in these findings. Another potential limitation is that this study did not consider the impact of extraneous variables that might impact the perceived negativity of the film montage, such as the use of self-regulatory strategies or thought challenging, which occur after exposure to the information but before the film montage. This is important, as such variables may inflate the perceived impact of expectancies on the degree to which the event was experienced as negative and overrepresent the protective value of expectancies against low emotional resilience in the wake of a stressful event. Future studies may wish to screen for such variables to determine the validity of these findings.

## 5. Conclusions

The present study investigated the contributions of negative expectancy bias to emotional resilience indexed across two domains: anxiety reactivity in the lead up to a potentially stressful event and the perseveration of anxiety following the event. The results revealed that negative expectancy bias predicts anxiety reactivity and indirectly predicts anxiety perseveration through the mediating influence of the perceived negativity of the event. While further research is required to discriminate the validity of the relationship between negative expectancy bias and emotional resilience in the wake of a potentially stressful event, these findings support the notion that expectancies offer protective value against low emotional resilience both in anticipation of and in the wake of potentially stressful events. These findings also have implications for clinical interventions, which may benefit from targeting negative expectancy bias as a means of improving emotional resilience.

## Figures and Tables

**Figure 1 behavsci-15-00531-f001:**
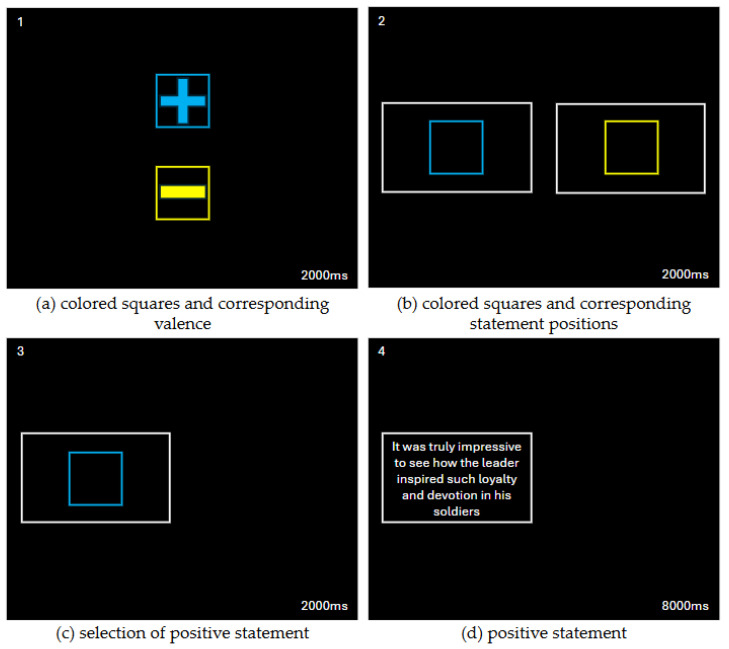
Diagram depicting the sequence of events for a single trial of the Information Statement Presentation and Selective Interrogation Bias Assessment.

**Figure 2 behavsci-15-00531-f002:**
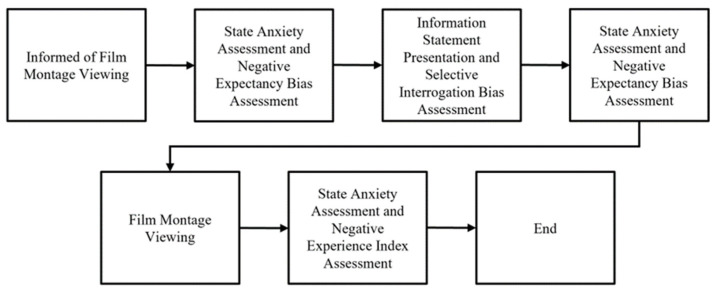
Diagram depicting the sequence of events during the experiment.

**Table 1 behavsci-15-00531-t001:** Descriptive statistics for anxiety reactivity, anxiety perseveration, negative expectancy bias elevation, negative experience index, and negative selective interrogation bias.

	Mean	Median	SD	Minimum	Maximum
Anxiety Reactivity	44.4	27	96.2	−161	292
Anxiety Perseveration	331	330	132	45	587
Negative Expectancy Bias Elevation	18.9	8	79.3	−197	209
Negative Experience Index	93.3	92	134	−251	396
Negative Selective Interrogation Bias	0.45	0.48	0.21	0.02	1

## Data Availability

Information and resources relevant to the development of this study and data analysis are stored on the UWA Institutional Research Data Storage (IRDS).

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
