# Peer review of "The Contribution of Negative Expectancies to Emotional Resilience"

_behavsci, 2025, doi:10.3390/bs15040531_

Round 1

Reviewer 1 Report

Comments and Suggestions for Authors

The topic of emotional resilience in relation to negative expectancies represents a very interesting area of ​​research and very needed new insights in psychology science.

However, this manuscript needs some improvements.

The last reference: Tough, J., Grafton, B., MacLeod, C., & van Bockstaele, B. (under review). Role of Information Seeking in Anxiety-Linked Expectancies - needs to be excluded from this manuscript since it is under review, not published so cannot be read or content compared to this manuscript.

- It is not acceptable to base the large part of introduction as well as other parts of this manuscript on (or referring to) unpublished work.

- try to include that content in the manuscript in another way without referring to the manuscript under review.  Or alternatively, wait for the mentioned article to be published, then enter the complete reference.

- the introduction is largely based on the unpublished work of the same authors, and it is recommended to revise the introduction in such a way as to cite the results of several other published studies in the field or bring up controversies based on previous studies etc.

Some additional suggestions:

Ln 121:  instead of " individuals" write "students"

Ln 126-127: " While the anticipation of a fictitious event was sufficient to assess anxiety reactivity in Tough et al. (under review)" exclude reference to unpublished work. Describe the method as it was for the present study.

Results:

Variables Anxiety Reactivity and Negative Expectancy Bias Elevation, clearly significantly deviate from the normal distribution, and it is not justified to use raw data in the regression analysis. 

Please give the explanation for the use of parametric analysis on data that do not meet the prerequisites for the use of parametric statistical procedures.

Has a transformation (and which) been done for the purpose of normalizing the distribution of these variables?

Ln 472: separate concluding text in Conclusion section.

Ln 474-475:  "The results revealed that negative expectancy bias predicts anxiety reactivity, as was initially observed by Tough et al. (under review)."  This statement is not acceptable for the conclusion because it refers to an unavailable article. Conclusion needs to be rewritten to reflect the conclusions of the results presented in this research/manuscript.

Reviewer 2 Report

Comments and Suggestions for Authors

The authors investigated the relationships among anxiety reactivity, anxiety perseveration, negative expectancy bias, and perceived negativity of an event in a sample of undergraduate students who viewed a film montage with both negative and benign content. They found that negative expectancy predicted both anxiety reactivity and anxiety perseveration, and that the latter effect was mediated by the perceived negativity of the event (i.e. the film montage). I found the results interesting but felt the manuscript could be improved with some clarifications, which are outlined below.

-It would be helpful to specify the participant population in the abstract (i.e. healthy adults or undergraduate students).

-“Investigators have typically distinguished between emotional resilience in the lead up to a potentially stressful event (Lin et al., 2023), which reflects the degree to which negative emotion becomes inflated as one approaches a potentially stressful event”; Lin et al. doesn’t seem to define anxiety reactivity this way, but discusses resilience during rather than before a stressful event, e.g. “Resilience is traditionally described as the ability to resist or ‘bounce back’ from negative emotional experiences (Luthar et al., 2006) and adapt flexibly to stressful events (Lazarus, 1993; Puolakanaho et al., 2023).”

-“While it may be the case that the degree to which individuals expect potentially stressful events to be more negative, relative to positive, may underly emotional resilience manifest as both anxiety reactivity and anxiety perseveration, it is also possible that anxiety perseveration may be driven by different mechanisms”; This sentence is a bit hard to read.

-It might be helpful to include the scale items in appendix (for assessments that have not previously been published).

-“Participants’ average score on the Positive Expectancy Items was subtracted from their average score on the Negative Expectancy Items, resulting in a measure of Negative Expectancy Bias. This score represents the proportion of Negative Expectancy Items, compared to Positive Expectancy Items, that were endorsed by participants.” This seems to suggest that items were categorial (i.e. endorsed or not endorsed), while the previous sentence states that item scores were continuous and ranged from 1-100. It would be helpful to clarify this point.

-“State anxiety was assessed at multiple points as the film montage approached to allow for a measure of anxiety reactivity.” The wording here suggests that state anxiety was assessed more than twice, but the description below seems to indicate it was assessed twice as the montage approached (at the beginning of the experiment and immediately before the montage). The reference to multiple assessments is also present in the abstract.

-“To measure anxiety reactivity, state anxiety was assessed at the beginning of the experiment and immediately before viewing the film montage.” It would be helpful to specify here how much time elapsed between these two assessments.

-“State anxiety was also assessed after the film montage viewing, the scores from which were used as an index of Anxiety Perseveration.” Were baseline state anxiety scores subtracted here as was done for the anxiety reactivity measure? If not, why not? I.e., it would seem that anxiety perseveration would be conceptualized as persistence of elevated state anxiety after a stressful event, compared to baseline anxiety level.

-“These two information sets collectively comprised 32 statements, which respectively describe negative and benign information about each of the eight video clips in the film montage.” It’s stated earlier that 12 clips were included in the montage.

-“3.3.3. Effect Concerning Association Between Negative Selective Interrogation Bias and 322 Negative Expectancy Bias Elevation?” Use of a question mark here is inconsistent with other subheadings.

-“This indicates that the index of Negative Selective Interrogation Bias only impacts the index of Anxiety Reactivity through its impact on the index of Negative Expectancy Bias Elevation.” I think this is an overly strong causal inference based on the statistical mediation analysis.

-“As per the second aim of the current study, the following analyses were conducted 354 to investigate whether there is an association between negative expectancy bias and anx- 355 iety preservation”; There is a typo here (should be “perseveration”).

-“This indicates that Negative Expectancy Bias scores only impact 385 Anxiety Perseveration scores through their impact on Negative Experience Index scores.” Again, I think this suggests and overly strong causal interpretation.

Round 2

Reviewer 1 Report

Comments and Suggestions for Authors

Revised manuscript is acceptable for publication. No further requirements for improvnemts and changes.

Author Response

N/A